**Data Availability Statement:** Attached as Supporting Information

**Funding:** The authors received no specific funding for this work.

# Informal welders' occupational safety and environmental health risks in northwestern Tanzania

**Leah Magoha[1], Elias C. Nyanza**[2]*, **Moses Asori**[3], **Deborah S.K. Thomas**[3]

**1** Occupational Health and Safety Authority, Lake Zone Office, Mwanza, Tanzania, **2** Department of Environmental, Occupational Health, and GIS, School of Public Health, Catholic University of Health and Allied Sciences, Bugando, Mwanza, Tanzania, **3** Department of Geography & Earth Sciences, The University of North Carolina at Charlotte, Charlotte, NC, United States of America

* elcnyanza@gmail.com

## Abstract

Often with minimal formal training and protections, informal welders face significant occupational health and safety (OSH) risks. This cross-sectional study of 219 adult informal welders at 70 informal welding sites in Mwanza City, Tanzania aimed to: 1) capture knowledge and awareness of occupational risks and safety precautions, training, and self-reported work-related injuries and illness and 2) observe worker use of personal protective equipment and site safety. We hypothesized that knowledge, awareness, and site inspections would improve use of PPE and that improved safety and site inspections would reduce self-reported injuries and illness. A generalized linear model (GLM) was used to model all relationships. Robust standard error estimation was used to avoid overestimation of parameters. Having a post-secondary education (a$\beta$ = 1.01, 95% CI: 0.962, 1.061; $p$ = 0.0679), having training in OSH (a$\beta$ = 0.927, 95% CI: 0.872, 0.984, $p$ = 0.014), increased knowledge of occupational risks (a$\beta$ = 1.305, 95% CI: 1.143, 1.491; $p$<0.001), and knowledge of safety measures (a$\beta$ = 1.112, 95% CI: 0.881, 1.404; $p$ = 0.372) increased PPE use by 1%, 7.3%, 30.5%, and 11.2% respectively. Workers who used PPE were less likely to experience fire explosions (AOR = 0.149, 95% CI: 0.029, 0.751; $p$ = 0.02), radiation exposure (AOR = 0.097, 95% CI: 0.016, 0.579, $p$ = 0.01) or electric shocks (AOR = 0.012; 95% CI: 0.001, 0.11, $p$<0.001). Having increased knowledge of safety practices also decreased the odds of fire explosions (AOR = 0.075, 95% CI: 0.018, 0.314; $p$<0.001). Those with higher knowledge of occupational risk (a$\beta$ = 1.57, 95% CI: 1.404, 1.756; $p$<0.001) and safety measures (a$\beta$ = 1.628, 95% CI: 1.34, 1.978; $p$<0.001) were more likely to have more positive attitudes towards safety practices. Our findings suggest that comprehensive targeted interventions including increased knowledge of occupational risks, safety practices, and occupational health law through training, along with enforcement and inspection by government officials, would benefit the environmental and occupational health for informal welders.

**Competing interests:** The authors have declared that no competing interests exist.

## 1. Introduction

In Africa, an estimated 71.9% of non-agriculture employment is informal. Often with minimal formal training and protections, workers face significant occupational health and safety risks with approximately 2 million people perishing every year from work-related causes globally [1, 2]. In Southern Africa, an estimated 18,000 people die from work-related illnesses every year and 13 million are injured and 67,000 contract work-related illnesses [3]. In recent years, Tanzania has experienced increased development, including large-and small-scale industries, building, and manufacturing, requiring welding. Most welders in Tanzania work in the informal sector with limited adoption of safety precautions [3–5].

The welding process joins and cuts metal using a flame or an electric arc and other heat sources to melt and cut or to melt and join two pieces of metal [6]. There are several types of arc welding, including stick or shielded metal arc welding (SMAW), the gas-shielded methods of metal inert gas (MIG), tungsten inert gas (TIG), plasma arc welding (PAW), and submerged arc welding (SAW) [7, 8]. The most common welding method used by informal welders is the SMAW, which uses electricity as a source of heat [9, 10]. Informal welding sites, like many other small-scale industries, are characterized by low capitalization, and minimum investment in safety precaution procedures, putting those involved and the communities around them at risk [6].

Welding is a hazardous profession with many health and safety risks to workers, including heat exposure, burns, dangerous flying objects, radiation exposure, noise pollution, fumes (for example particulate matter - $PM_{2.5}$ and $PM_{10}$), electrocution, and poor ergo-dynamics [6, 11, 12]. These occupational hazards can result in temporary and/or permanent injury, short- or long-term adverse health effects, discomfort, and even death to workers [6, 11]. Short-term effects include cuts and burns, irritation of the eyes, nose, chest, and respiratory tract, muscle aches, coughing, fatigue, and nausea. Long-term health effects include an increased risk of pulmonary siderosis due to exposure to iron dust, lung, larynx, and potential cancers from agents found in smoke [6, 12]. Many of these hazards can be minimized with adherence to occupational health and safety guideline, using personal protective equipment (PPE), maintaining equipment, practicing electrical safety, and employing proper hazard control measures [6, 9–11, 13]. Welders should not weld or cut unless wearing the necessary PPE to protect all areas of their body from injury during welding or cutting [14]. Proper welding PPE consists of a welding hood with a face shield, a respirator or mask, safety glasses, earplugs, gloves, long-sleeved jacket, closed-toed shoes, and ear plugs [14].

The Tanzanian Occupational Health and Safety Authority (OSHA) regulates and oversees all occupational health and safety (OHS) across the Tanzania mainland, as stipulated by the *OHS Act Number 5 of 2003* and the *OHS Policy of 2009* [15, 16], aiming to promote and maintain physical, mental, and social well-being of workers across occupations [14]. However, the Tanzanian laws and policies are not as comprehensive as they could be to protect a broad population. Further, safety service provision and inspection from the government is scarce [3]. Typically, adherence to various OHS guidelines is primarily the responsibility of workers and their managers at workplaces. Yet, knowledge of, and adherence to, workplace safety practices and guidelines, remains a significant challenge [3]. As such, coupled with limited site inspection, safety practices informed by the laws and practices is often quite low among welders [17].

In parallel to rapid population growth across Tanzania from about 44.9 million in 2012 to 61.7 million in 2022 [18], Mwanza City, the third largest city in Tanzania and the largest in the Lake Zone along Lake Victoria, has also increased. This substantial population growth has spurred building and development with a significant increase in the number of small-scale industries with a corresponding increase in welding activities. For example, in 2017, the

*Mwanza Regional Investment Guide* estimated that approximately 13% of the total small-scale industries were involved in welding activities [19].

Although welding activities employ a significant number of people in Tanzania, there is limited information on the welders' knowledge of, and adherence to, safety practices as outlined in Tanzanian OSH and how this affects workers' exposure to occupational health and hazards. Evolving evidence from other African countries reveals limited knowledge about and use of PPEs among welders in the workplace. For example, a study conducted in Kaduna, Nigeria, found that only 34.2% of welders used one or more of PPE during welding activities [20]. Another study from Ethiopia found that only 32.3% of welders were knowledgeable of occupational safety practices [21], whereas 62.6% of welders in Delta State reported awareness of OHS guidelines [22]. In Sokoto in Nigeria, 9.6% to 83.2% of welders never used PPE at all [23].

In addition to the low level of knowledge and limited use of PPE, informal welders do not benefit from training programs or oversight through regular OSHA and/or health officers from various municipals inspections. Most informal sector workers do not have formal training and lack access to training opportunities at the welding site. There is simultaneously a lack of comprehensive surveillance of occupational hazards and/or injuries in the informal sector and so the degree of workers environmental and occupational health risk is not well documented.

To address this gap, our study of informal welders evaluated the relationship of OHS knowledge, training, and site inspection to health and safety guidelines adherence, including the use of PPE. This cross-sectional study of 219 adult informal welders at 70 informal welding sites in Mwanza City, Tanzania aimed to: 1) capture knowledge and awareness of occupational risks and safety precautions, training, and self-reported work-related injuries and illness and 2) observe worker use of personal protective equipment and site safety. We hypothesized that knowledge, awareness, and site inspections would improve use of PPE and that improved safety and site inspections would reduce self-reported injuries and illness.

## 2. Materials and methods

### 2.1. Study design and setting

The cross-sectional study surveyed 219 adult (older than 18 years of age) informal welders working at 70 informal electric welding sites across Mwanza City, in northwestern Tanzania on Lake Victoria from June 2021 –May 2022. Mwanza City is comprised of two main administrative districts: Nyamagana District (with 12 wards) and Ilemela District (with 9 wards). According to the 2012 national population census, Mwanza city had a population of 706,453 (342,530 male and 363,923 female) [18, 24].

### 2.2. Sampling and recruitment

The Mwanza City Council (MCC) of Social Affairs maintains an inventory of groups of welders who register for social affairs benefits, including getting loan support from the government. They do not necessarily register as a site (though it is possible) and welders do not register as individuals. The group is formal, but individuals who are informal workers cannot apply directly as an individual. This functions to access capital since informal workers cannot typically access formal bank loans due to extensive requirements for documentation, including financial stability. The organized group that registers determines what the loan request is, for example improving the welding site, and the group takes social accountability.

Using the inventory of registered welding groups, areas with the highest concentrations of welding activities were identified to target sampling. Then, within each of these areas, sites were identified by walking along the street and stopping at each welding site. All welders who

were present at that time at a specific site were invited to participate. Participation was voluntary; a total of 219 adult welders signed the consent forms and participated in the study.

## 2.3. Data collection

A semi-structured questionnaire was adapted from a previous study that examined prevalence and determinants of occupational injuries among welders in small scale metal workshops in Wakiso District, Uganda (2021) and incorporating information from the *Tanzanian OSHA Fifth Schedule, sections 63,62,61, and 95 of OSH Act no 5 of 2003 and 2015 rules* [15, 16]) (*See S1 Table*). The survey also collected information on socio-demographic characteristics, including education level, history of working on welding sites, working hours, and age. Respondents answered a range of questions about, 1) knowledge of existing safety regulatory guidelines, 2) knowledge of occupational safety practices, 3) knowledge of occupational risks, and 4) perceived relevance of occupational safety practices (PRSP). Individual welders were asked about the area of the welding process and site where they worked for more than 90% of their time at the welding site. Composite scores were created for occupational risk, (three yes/no questions), knowledge of occupational safety practices (10 questions, very important/important/not important), PRSP (four questions, relevant/not relevant) (*See S2 Table*). A score for a given category was computed as follows:

$$Composite_X = \frac{\sum N_{X_i}}{\sum_{i=1}^{K} N_X} \tag{1}$$

where $\sum N_{X_i}$ is the sum of the number of questions with response given by participant *i*, whereas $\sum_{i=1}^{K} N_X$ is the total number of questions for a given category (e.g. PRSP) considered in our checklists. No composite value was computed for awareness of occupational regulatory guidelines, as this category had one question with a binary response.

To enhance quality of the results, the survey was piloted to ten welders outside of the study area to inform the refinement of the survey and ensure questions were clear to the respondent. The questionnaire was administered orally in-person with responses recorded on paper during a face-to-face interview at the job site. In addition, a pre-designed checklist was used by the data collector to capture information on adherence to OSH guidelines. The inspection checklist, adapted from the *Tanzanian OSHA Fifth Schedule, sections 63,62,61, and 95 of OSH Act no 5 of 2003 and 2015 rules* [15, 16], captured workers' use of PPE (safety boots, safety goggle, respiratory or dust mask, and gloves) during welding activities based on the type of activity and associated required PPE as specified by the American Safety and Health Fact Sheet for Welding and Cutting [14]. The checklist also captured site hazards, including fire risks, lack of grounding, water on-site, sanitary conveniences, washing facilities, lack of first aid facilities, protection against dust and fumes and protection of eyes in welding (*see S3 Table*). Binary response for each PPE use: yes = 1; no = 0. The composite score for the PPE using the same equation as for the survey questions. The use of equipment was recorded as yes/no, as was the number of times a safety violation was observed.

## 2.4. Ethical considerations

Ethical clearance was obtained from the joint Catholic University of Health and Allied Science (CUHAS) and Bugando Medical Center Ethics and Research Review Committee (Certificate no *CREC/476/2021*). Permission to conduct the research was obtained from the Mwanza Regional Administrative Secretary Office (*Ref. NO. FA.137/264/01J/23*), Mwanza City Council (*Ref. No. T40/7/122*) and Ilemela District Council (*Ref. IMC/T.40/7/VIL.IV/15*). Written

informed consent was obtained from all informal welders who participated in this study. Participation in this study was voluntary. Participants signed an informed consent form (written in Kiswahili, the primary language of most of the population of Tanzania) prior to data collection (*See S1 Checklist*). In instances where individual welders had low literacy, a research assistant read and reviewed the consent form with the prospecting participants and a thumbprint was obtained indicating their consent. In all cases, an impartial witness from the local community was present during the informed consent process.

## 2.5. Statistical analysis and data management

Data cleaning and analysis were performed using SPSS Statistics 23 (SPSS Inc., Chicago IL) and RStudio version 4.2.1. A preliminary exploration of the data was done to check for missing values, duplicates, and unusual observations before analysis. In descriptive statistics, continuous variables were summarized using median and interquartile range while categorical variables were summarized using frequency and percentages.

A generalized linear model (GLM) was used for three separate analyses. The first assessed the association between knowledge of risks, knowledge of safety measures, site inspection, training in OHS, and educational level (predictors), and the use of PPE (response variable). Next, the impacts of knowledge of safety, knowledge of risk and PPE were used to predict the odds of occupational hazards. Since occupational hazards, including fire explosions, radiation exposure, electric shocks, and head/hand cuts were all binary outcomes, the binary logistic function in GLM was used to compute the odds. Lastly, the association between knowledge of risks and safety measures, site inspection, age, duration at the site, educational level (predictors), and welders' PRSP (response variable) was examined. Association was considered statistically significant at $p < 0.05$.

## 3. Results

### 3.1. Descriptive statistics

Table 1 and Fig 1 summarizes the background and characteristics of informal welders, along with the composite scores for knowledge, safety, and adherence (*See S1 Data* for extended data information). All of 219 informal welders were men and nearly half (49%) were aged between 25–34 years (median 29, IQR 25–34). The median duration of working at the site was 4 years (IQR: 2–6 years). Most (97.3%) of the informal welders had completed at least primary education and 10.9% knew of OSH laws and regulations. Although 57.1% of the informal welders reported having at least one OSH training, only 11.4% reported being inspected by government authorities on-site (Fig 1). Regarding training, 32% of welders had formal vocational and technical training, whereas 68% received their training onsite. Only 37% of the informal welders had knowledge of occupational safety practices and 93.6% had poor adherence to OSH guidelines. A majority (69%) had a high perception of PRSP. The median knowledge of occupational risk score was 3 (IQR 2–3), knowledge of safety measures 17 (IQR 16–18), PPE use score 5 (IQR 4–5) and PSPP 4 (IQR 3–4).

Reported injury among participants reveals a significant occupational health burden. A large percentage of the participants reported experiencing electric shock (61.64%; n = 35), fire outbreaks (57.9%; n = 49), respiratory complications (64.8%; n = 142), and nasal problems (73.5% n = 161).

**Table 1. Background and characteristics of the informal welders (N = 219).**

| Variables | n | % |
|---|---|---|
| **Age** | | |
| 19–24 | 72 | 32.88 |
| 25–34 | 107 | 48.86 |
| 35–69 | 40 | 18.26 |
| Median age (IQR) | 28 (24–32) | |
| **Education level** | | |
| No formal education | 6 | 2.74 |
| At least primary education | 213 | 97.26 |
| **District** | | |
| Nyamagana | 38 | 17.35 |
| Ilemela | 181 | 82.65 |
| **Jobs Involved**\*\* | | |
| Welding only | 73 | 33.33 |
| Cutting and hammering only | 34 | 15.53 |
| Painting only | 12 | 5.48 |
| Involved in all activities | 100 | 45.66 |
| **Working experience** | | |
| Less 1 year | 47 | 21.46 |
| 1–5 years | 100 | 45.66 |
| Above 5 years | 72 | 32.88 |
| Median duration at the site (IQR) | 4(2–6) | |
| **Received any training** | | |
| The Vocational Training Center (VTC) | 71 | 32.42 |
| Site Training | 148 | 67.58 |
| **Safety training gained** | | |
| At least one safety training | 125 | 57.08 |
| None | 94 | 42.92 |
| **Inspection from gov't authorities** | | |
| Yes | 25 | 11.42 |
| No | 194 | 88.58 |
| **Knowledge of OHS laws** | | |
| Yes | 24 | 10.96 |
| No | 195 | 89.04 |
| **Safety practices knowledge rating** | | |
| Has knowledge (50% and above) | 81 | 36.99 |
| Do not have knowledge (<50%) | 138 | 63.01 |
| **Perceived safety practice status** | | |
| Good | 150 | 68.49 |
| Poor | 69 | 31.51 |
| **Adherence to OHS guidelines** | | |
| Good (50% and above) | 14 | 6.39 |
| Poor (<50%) | 205 | 93.61 |

Note: \*\**The most common activities or job reported to be carried out by respondents by 90% of their time at the welding workshop.*

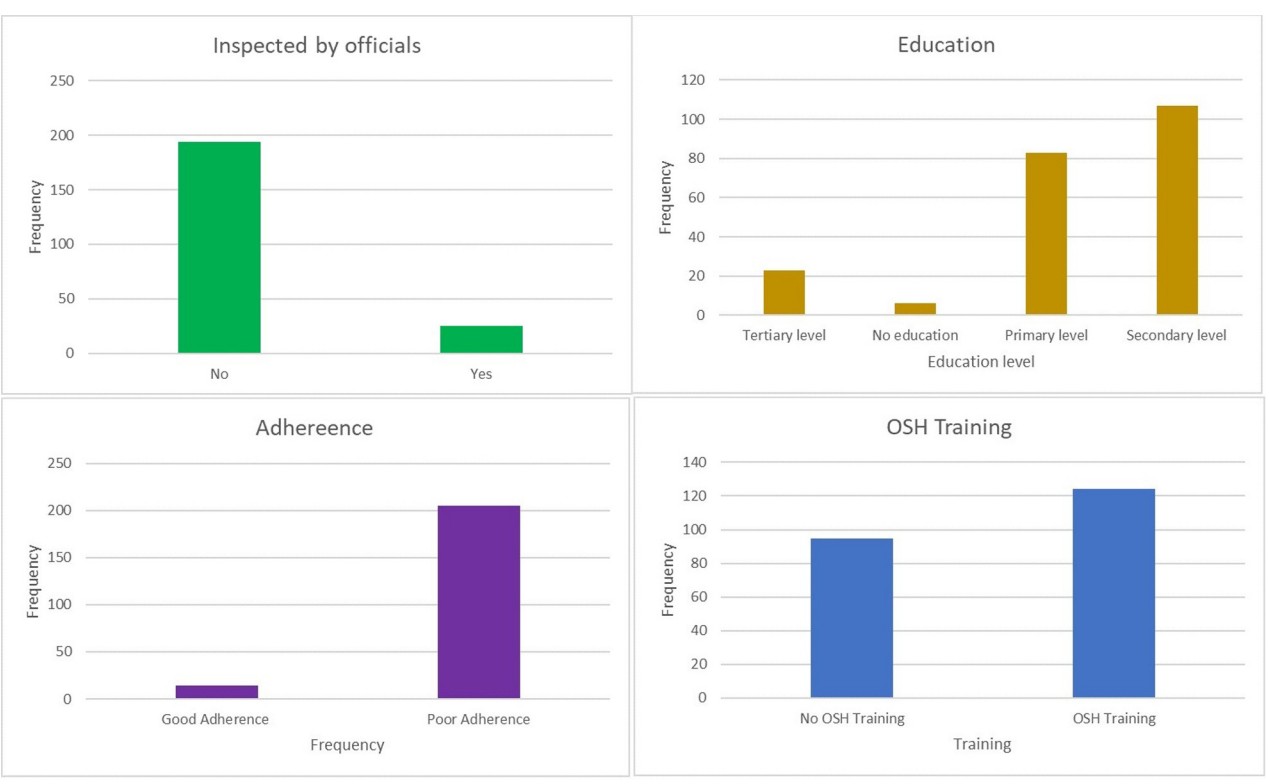

**Fig 1. Characteristics of informal welders (*OSH training, adherence, inspection and education*).**

### 3.2. Factors associated with PPE use

In the unadjusted model, knowledge of occupational risks (c$\beta$ = 1.335; 95% CI: 1.184, 1.506, $p$<0.001), site inspection (c$\beta$ = 1.103; 95% CI: 1.003, 1.212, $p$ = 0.043) and having training in OSH (c$\beta$ = 1.089; 95% CI: 1.025, 1.157, $p$ = 0.006) were significantly positively associated with the use of PPE, while knowledge of safety measures (c$\beta$ = 1.149; 95% CI: 0.896, 1.473, $p$ = 0.273) was not statistically significant, even though it increased PPE use by 14%. Having primary (c$\beta$ = 0.871; 95% CI: 0.762-.995, $p$ = 0.042) and secondary-level education (c$\beta$ = 0.945, 95% CI: 0.830–1.075, $p$ = 0.758) were associated with less use of PPE, even though the relationship was not statistically significant for secondary school. Furthermore, duration working at the site (c$\beta$ = 1.003; 95% CI: 0.996, 1.011, $p$ = 0.428) and age (c$\beta$ = 1.000; 95% CI: 0.996, 1.005, $p$ = 0.809) were not significantly associated with PPE use (Table 2).

In our adjusted model, an increased knowledge of occupational risks was associated with a 31% increased use of PPE (a$\beta$ = 1.305, 95% CI: 1.143, 1.491; $p$<0.001). Education had no association with PPE use. Increased knowledge of occupational safety measures increased the use of PPE by 11% (a$\beta$ = 1.112, 95% CI: 0.881, 1.404; $p$ = 0.372) even though the association was also not statistically significant. Additionally, we found that = those with training had a 7.5% increase in PPE use (a$\beta$ = 1.0753, 95% CI: 1.147–1.016, $p$ = 0.029) over those with no training. Further, sites with no government safety inspection had a 7.2% decline in use of PPE (a$\beta$ = 0.928; 95% CI: 0.872, 0.988, $p$ = 0.019).

**Table 2. Crude and adjusted model for (1) factors associated with PPE use, (2) factors associated with PSPP and (3) associated factors for occupational hazards.**

| | Crude Model | | | Adjusted Model | | |
|---|---|---|---|---|---|---|
| Variable | Coefficient | 95% CI (LL-UL) | *p*-value | Coefficient | 95% CI (LL-UL) | *p*-value |
| *Factors associated with PPE use.* | | | | | | |
| Knowledge of occupational risks | 1.335 | 1.184–1.506 | <0.001*** | 1.305 | 1.143–1.491 | <0.001*** |
| Site inspection | 1.103 | 1.003–1.212 | 0.043* | 0.928 | 0.872–0.988 | 0.019* |
| Having training in OSH | 1.089 | 1.025–1.157 | 0.006** | 1.0753 | 1.147–1.016 | 0.029* |
| Knowledge of safety measures | 1.149 | 0.896–1.473 | 0.273 | 1.112 | 0.881–1.404 | 0.372 |
| **Education** | | | | | | |
| Post-secondary education | 0.962 | 0.825–1.120 | 0.616 | 0.942 | 0.788–1.127 | 0.514 |
| Secondary-level education | 0.945 | 0.830–1.075 | 0.387 | 0.925 | 0.792–1.082 | 0.331 |
| Having primary | 0.871 | 0.762-.995 | .042* | 0.887 | 0.758–1.040 | 0.139 |
| Having no education | Ref | ---- | ---- | ref | ------ | ------ |
| Duration working at the site | 1.003 | 0.996–1.011 | 0.428 | 1.008 | 0.999–1.017 | 0.066 |
| Age of Welders | 1.000 | 0.996–1.005 | 0.809 | 0.997 | 0.993–1.000 | 0.042* |
| *Factors associated with perceived relevance of safety practices (PRSP)* | | | | | | |
| Knowledge of occupational risks | 1.865 | 1.416, 2.472 | 0.001** | 1.574 | 1.404, 1.756 | <0.001** |
| Knowledge of occupational safety | 1.657 | 1.475, 1.861 | 0.001** | 1.628 | 1.34, 1.978 | <0.001** |
| Site inspection (Yes is ref) | 0.953 | 0.87, 1.037 | 0.245 | 0.994 | 0.923, 1.070 | 0.873 |
| Age of welders | 1.006 | 1.003, 1.009 | <0.001** | 1.001 | 0.998, 1.004 | 0.527 |
| Duration at site | 1.01 | 1.004, 1.015 | <0.001** | 1.003 | 0.998, 1.008 | 0.226 |
| **Education** | | | | | | |
| Post-secondary education | 1.199 | 0.991, 1.449 | 0.065* | 1.195 | 1.023, 1.397 | 0.025* |
| Secondary-level education | 1.202 | 1.010, 1.430 | 0.038* | 1.187 | 1.031, 1.368 | 0.017* |
| Having primary | 1.147 | 0.963, 1.367 | 0.124 | 1.179 | 1.021, 1.361 | 0.025* |
| Having no education | Ref | ---- | ----- | --- | ---- | ---- |
| *Risk factors of occupational hazards* | | | | | | |
| head cuts | | | | | | |
| Coefficient (OR) | | | | | | |
| PPE use | 0.038 | 0.008, 0.179 | 0.001** | 0.052 | 0.010, 0.279 | <0.001** |
| PRSP | 0.067 | 0.01, 0.299 | <0.001** | 0.761 | 0.565, 1.026 | 0.073 |
| Knowledge of occupational risks | 0.157 | 0.044, 0.566 | 0.005** | 0.577 | 0.120, 2.771 | 0.492 |
| Knowledge of safety measures | 0.002 | 0.000, 0.068 | <0.001** | 0.009 | 0.000, 0.267 | 0.006** |
| Electric shocks | | | | | | |
| PPE use | 0.042 | 0.007, 0.275 | 0.001** | 0.019 | 0.002, 0.164 | <0.001** |
| PRSP | 0.756 | 0.056, 10.273 | 0.833 | 0.362 | 0.078, 1.673 | 0.193 |
| Knowledge of occupational risks | 0.328 | 0.024, 4.570 | 0.407 | 0.076 | 0.002, 2.583 | 0.152 |
| Knowledge of safety measures | 0.065 | 0.006, 0.770 | 0.03* | 0.548 | 0.002, 197.126 | 0.841 |
| fire explosions | | | | | | |
| PPE use | 0.060 | 0.015, 0.246 | <0.001** | 0.105 | 0.023, 0.475 | 0.003** |
| PRSP | 0.043 | 0.009, 0.199 | <0.001** | 0.075 | 0.018, 0.314 | <0.001** |
| Knowledge of occupational risks | 0.107 | 0.029, 0.395 | <0.001** | 0.548 | 0.102, 2.936 | 0.483 |
| Knowledge of safety measures | 0.065 | 0.006, 0.770 | 0.03* | 0.215 | 0.012, 3.748 | 0.292 |
| Radiation exposure | | | | | | |
| PPE use | 0.089 | 0.019, 0.426 | 0.002** | 0.127 | 0.023, 0.708 | 0.019* |
| PRSP | 0.216 | 0.022, 2.078 | 0.185 | 0.738 | 0.136,0.934 | 0.040* |
| Knowledge of occupational risks | 0.107 | 0.029, 0.395 | <0.001** | 0.263 | 0.037, 1.860 | 0.181 |
| Knowledge of safety measures | 0.065 | 0.006, 0.770 | 0.03* | 1.260 | 0.031, 50.931 | 0.902 |
| Hand cuts | | | | | | |

*(Continued)*

**Table 2.** (Continued)

| Variable | Crude Model | | | Adjusted Model | | |
|---|---|---|---|---|---|---|
| | Coefficient | 95% CI (LL-UL) | *p*-value | Coefficient | 95% CI (LL-UL) | *p*-value |
| PPE use | 0.601 | 0.121, 3.034 | 0.543 | 0.747 | 0.138, 4.052 | 0.735 |
| PRSP | 0.257 | 0.049, 1.346 | 0.108 | 0.783 | 0.625, 0.98 | 0.034* |
| Knowledge of occupational risks | 0.673 | 0.144, 3.146 | 0.615 | 0.608 | 0.08, 4.440 | 0.624 |
| Knowledge of safety measures | 0.223 | 0.014, 3.525 | 0.287 | 0.407 | 0.018, 9.166 | 0.572 |

### 3.3. Factors associated with perceived relevance of safety practices (PRSP)

In the unadjusted model, we found that increased knowledge of occupational risks was associated with an 87% (c$\beta$ = 1.865, 95% CI: 1.416, 2.472, *p* = 0.001) increase in PRSP and an increased knowledge of occupational safety measures was associated with a 66% (c$\beta$ = 1.657, 95% CI: 1.475, 1.861; *p*<0.001) increase in PRSP. Those sites with no government safety inspection had a 5% reduction in PRSP (c$\beta$ = 0.953, 95% CI: 0.87, 1.037, *p* = 0.245) even though the association was not statistically significant. Furthermore, age (c$\beta$ = 1.006, 95 CI: 1.003, 1.009; *p*<0.001) and longer working duration at the welding sites (c$\beta$ = 1.01, 95% CI: 1.004, 1.015; *p*<0.001) both increased PRSP by 0.6% and 10% respectively. Additionally, as compared to having no education, having primary (c$\beta$ = 1.147; 95% CI: 0.963, 1.367, *p* = 0.124), secondary (c$\beta$ = 1.202; 95% CI: 1.010, 1.430, *p* = 0.038) and post-secondary education (c$\beta$ = 1.199; 95% CI: 0.991, 1.449, *p* = 0.062) was positively associated with PRSP.

In the adjusted model, increased knowledge of occupational risks was linked with a 57% (a$\beta$ = 1.57, 95% CI: 1.404, 1.756; *p*<0.001) increase in PRSP. A higher knowledge of occupational safety measures was associated with a 63% (a$\beta$ = 1.628, 95% CI: 1.34, 1.978; *p*<0.001) increase in PRSP. However, age 0.1% (a$\beta$ = 1.001, 95% CI: 0.998, 1.004; *p* = 0.527) and duration at site 0.3% (a$\beta$ = 1.003, 95% CI: 0.998, 1.008; *p* = 0.226) were not statistically significantly associated with PRSP. Lastly, as compared with having no education, having at least primary (a$\beta$ = 1.142; 95% CI: 0.99, 1.310, *p* = 0.059), secondary (a$\beta$ = 1.159; 95% CI: 1.012, 1.327, *p* = 0.033) and college level education (a$\beta$ = 1.172; 95% CI: 1.010, 1.359, *p* = 0.036) was associated with higher PRSP.

### 3.4. Risk factors of occupational hazards

In the crude model, PPE use was statistically significantly associated with reduced odds of head cuts (COR = 0.038; 95% CI: 0.008, 0.179, *p*<0.001), electric shocks (COR = 0.042; 95% CI: 0.007, 0.275, *p*<0.001), fire explosions (COR = 0.060; 95% CI: 0.015, 0.246, *p*<0.001), and radiation exposure (COR = 0.089; 95% CI: 0.019, 0.426, *p* = 0.002), but not significantly associated with hand cuts (COR = 0.60; 95% CI: 0.121, 3.034, *p* = 0.543). Furthermore, higher PRSP was associated with decreased odds of electric shocks (COR = 0.756; 95% CI: 0.056, 10.273, *p* = 0.833), radiation exposure (COR = 0.216; 95% CI: 0.022, 2.078, *p* = 0.185), fire explosions (COR = 0.043; 95% CI: 0.009, 0.199, *p*<0.001), hand cuts (COR = 0.257; 95% CI: 0.049, 1.346, *p* = 0.108) and head cuts (COR = 0.067; 95% CI: 0.01, 0.299, *p*<0.001), even though none was statistically significant. Additionally, increase knowledge of occupational risks was associated with reduced odds of hand cuts (COR = 0.673, 95% CI: 0.144, 3.146, *p* = 0.615), head cuts (COR = 0.157; 95% CI: 0.044, 0.566, *p* = 0.005), electric shocks (COR = 0.328; 95% CI: 0.024, 4.570, *p* = 0.407), fire explosions (COR = 0.107; 95% CI: 0.029, 0.395, *P*<0.001), and radiation exposure (COR = 0.29, 95% CI: 0.064, 1.377, *p* = 0.121), even though only fire explosions was statistically significant. Increased knowledge of safety measures was significantly associated

with lesser odds of fire explosions (COR = 0.065; 95% CI: 0.006, 0.770, *p* = 0.03) and head cuts (COR = 0.002; 95% CI: 0.000, 0.068, *p*<0.001), but was not significant for electric shocks (COR = 1.086; 95% CI: 0.012, 96.096, *p* = 0.97), radiation exposure (COR = 0.32; 95% CI: 0.015, 6.732, *p* = 0.463) and hand cuts (COR = 0.223; 95% C: 0.014, 3.525, *p* = 0.287).

In the adjusted model, the use of PPE was found to have a significant impact on the odds of occupational hazards. For example, PPE use was associated with a significant reduction in head cuts (AOR = 0.052; 95% CI: 0.010, 0.279, *p*<0.001), electric shocks (AOR = 0.019; 95% CI: 0.002, 0.164, *p*<0.001), fire explosions (AOR = 0.105; 95% CI: 0.023, 0.475, *p* = 0.003), and radiation exposure (AOR = 0.127; 95% CI: 0.023, 0.708, *p* = 0.019), but not significant with hand cuts (AOR = 0.747; 95% CI: 0.138, 4.052, *p* = 0.735). Increased PRSP was associated with a significantly lesser likelihood of experiencing electric shocks (AOR = 0.362, 95% CI: 0.078, 1.673; *p* = 0.193), radiation exposure (AOR = 0.738, 95% CI: 0.136,0.934, *p* = 0.04), fire explosion (AOR = 0.075, 95% CI: 0.018, 0.314; *p*<0.001), hand cuts (AOR = 0.783, 95% CI: 0.625, 0.98; *p* = 0.034) and marginally significant with head cuts (AOR = 0.761, 95% CI: 0.565, 1.026; *p* = 0.073). There was no association between knowledge of occupational risks and radiation exposure (AOR = 0.263, 95% CI: 0.037, 1.86, *p* = 0.181), electric shocks (AOR = 0.076; 95% CI: 0.002, 2.583, *p* = 0.152), fire explosions (AOR = 0.548; 95% CI: 0.102, 2.936, *p* = 0.483), hand cuts (AOR = 0.608; 95% CI: 0.08, 4.440, *p* = 0.624) and head cuts (AOR = 0.577; 95% CI: 0.120, 2.771, *p* = 0.492). Similarly, there was no association between knowledge of safety measures and hand cuts (AOR = 0.407; 95% CI: 0.018, 9.166, *p* = 0.572), head cuts (AOR = 0.009; 95% CI: 0.000, 0.267, *p* = 0.006), electric shocks (AOR = 0.548, 95% CI: 0.002, 197.126, *p* = 0.841), fire explosion (AOR = 0.215; 95% CI: 0.012, 3.748, *p* = 0.292), and radiation exposure (AOR = 1.260; 95% CI: 0.031, 50.931, *p* = 0.902).

## 4. Discussion

Workplace safety is integral to industrial prosperity, but unfortunately most informal welders are subjected to substantial occupational health risks. Especially for informal welders who know little about risks and safety practices or the benefits of the use of PPE, poor health outcomes and injury increase substantially [17]. Our current study revealed that many of the informal welders had poor knowledge of workplace safety practices and limited knowledge of OHS guidelines set by the Tanzanian OSHA. This is consistent with previous studies from Nigeria and Tanzania where 64.9% of self-employed electric arc welders had poor knowledge of OSH and 55.7% had limited knowledge of OSH rules and regulations respectively [25, 26]. Most (87.1%) of the welders in this study had no formal training on safety measures and 37% rarely used PPE when performing their duties, which is consistent with previous studies from Tanzania and Nigeria [25, 26].

Poor knowledge of safety practices can be attributed to the modality of training. For example, most of the respondents in our study received their training informally through onsite training (e.g., via apprenticeship) and so rely extensively on their co-workers, who are likely untrained as well, for information regarding safety practices and risks associated with welding activities. However, contrary to our findings, studies from Jinja-Uganda and Ethiopia noted that 50% and 56% of informal welders were knowledgeable about safety practices respectively [21, 27]. The differences in the level of knowledge regarding safety practices between our findings and those from Kenya and Uganda may be attributed to variations underlying the socio-demographic backgrounds of the study subjects (e.g., education level) or possibly the (in) effectiveness of OSH guidelines regulation.

Research from Kenya and India reported a significant association between increased knowledge of occupational risks, knowledge of safety practices, degree of OSH law

enforcement, and level of education with the adoption of safety practices including the use of PPE [28, 29]. Our study is in alignment with these studies and further substantiates the need for formal training programs. Further, the importance of inspections from government oversight to enforce OSH, using this as an opportunity to increase awareness, substantially improves PPE use and decreases injuries.

The informal nature of training and almost no inspection from enforcement officials contributes to the majority (94%) of welders not adhering to the established OSH guidelines and by extension less use of PPE. This is also supported by a study from Delta State in Nigeria where increased use of PPE (62.6%) was associated with adherence to the OSH guidelines due to constant inspection at welding sites [22]. Crucially, although the majority (69%) of the informal welders had a positive perception regarding the relevance of safety practices, observing these practices, including the use of PPE, was more challenging for them. As another key finding in our study, higher levels of occupational risks/safety knowledge and heightened perception of importance of safety practices led to higher use of PPE and resulted in reduced injury. Welders who use PPE were less likely to experience fire explosions, radiation exposures, hand cuts, head cuts, and electric shocks. Providing the needed incentive, creating more awareness about occupational health, along with enforcement/inspection, would go a long way to make significant improvements in adopting safety practices.

Without the proper use of PPE, welders may be exposed to respirable dust which has the potential to penetrate the alveoli and result in serious respiratory health problems [30]. $PM_{2.5}$, or respirable dust, can pass through the nasal passage and penetrate the alveoli and may result in serious respiratory health effects depending on the content of the welding fumes [30, 31]. Further, welding fumes contain toxic chemicals, such as cadmium, chromium, manganese, and nickel from the base metals used in welding. These have been associated with lung cancer and neurological effects, especially upon prolonged exposure [32, 33]. A previous study in Dar es Salaam, Tanzania reported an association between metal fume ($PM_{2.5}$ 6.57±2.50mg/m$^3$) and respiratory symptoms among small-scale welders [9]. Prolonged exposure can result in serious chronic and debilitating respiratory health conditions. Taken together, the health and safety of informal welders can be improved immensely by proper use of PPE; training, education, policies, and site inspections together increase adherence.

## Study limitations

Due to COVID-19 restrictions and the need for adequate social distancing, we could not conduct a lung functioning test to understand the effects of air pollution, nor did we assess noise-induced hearing loss. Further, health effects and symptoms were self-reported and did not have a confirmed diagnosis. We did not test for the chemical composition of welding fumes, like cadmium, chromium, manganese, and nickel, or consider potential health outcomes from these exposures.

## 5. Conclusion

Evidence shows that increasing OSH education campaigns, enforcing (encouraging) welders to use PPE and safety practices, inspections from government officials, and providing the right incentive can improve the working conditions for informal welders. Less knowledge about safety practices contributes to reduced usage of PPE, which in turn results in more workplace injury. Continued worker health training programs that aim to increase knowledge of welding occupational risk and safety practices, refinement, and enforcement of OSH to directly address welding, and inspection by government officials will improve the health and safety for

informal welders. Coordinated intervention efforts between the Tanzanian government, academia, NGOs, and the welding site managers can have extensive worker health benefits.

## Supporting information

**S1 Checklist. Inclusivity in global research.**
(DOCX)

**S1 Data. De-identified data and detailed information regarding study participants.**
(CSV)

**S1 Table. Questionnaires on Assessment of informal welder's knowledge on safety practices and adherence to occupational health and safety guidelines.**
(DOCX)

**S2 Table. Composite variables used for constructing perceived safety practices (response variables), knowledge of safety practices, and risk scores (predictors).**
(DOCX)

**S3 Table. Checklist on adherence to OHS Guidelines requirements as per OHS ACT 5 OF 2003; and its regulations.**
(DOCX)

## Acknowledgments

The authors would like to acknowledge the Department of Environmental and Occupational Health at the Catholic University of Health and Allied Sciences, the Regional, District, and Local Authorities in Mwanza City for their collaboration and assistance in the completion of this study. The authors would like to thank all welders in Mwanza City for participating in this study.

## Author Contributions

**Conceptualization:** Leah Magoha, Elias C. Nyanza, Deborah S.K. Thomas.

**Data curation:** Leah Magoha, Elias C. Nyanza, Moses Asori, Deborah S.K. Thomas.

**Formal analysis:** Leah Magoha, Elias C. Nyanza, Moses Asori, Deborah S.K. Thomas.

**Investigation:** Leah Magoha, Elias C. Nyanza, Deborah S.K. Thomas.

**Methodology:** Leah Magoha, Elias C. Nyanza, Deborah S.K. Thomas.

**Project administration:** Leah Magoha, Elias C. Nyanza.

**Resources:** Elias C. Nyanza, Deborah S.K. Thomas.

**Software:** Elias C. Nyanza, Moses Asori.

**Supervision:** Elias C. Nyanza.

**Validation:** Leah Magoha, Elias C. Nyanza, Moses Asori, Deborah S.K. Thomas.

**Visualization:** Moses Asori, Deborah S.K. Thomas.

**Writing – original draft:** Leah Magoha, Elias C. Nyanza, Moses Asori, Deborah S.K. Thomas.

**Writing – review & editing:** Elias C. Nyanza, Deborah S.K. Thomas.

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
