## [Decision Letter · Decision Letter 0]

6 Sep 2023

PGPH-D-23-01493

Informal welders’ occupation safety and environmental health risks in northwestern Tanzania

Dear Dr. Nyanza,

Thank you for submitting your manuscript to PLOS Global Public Health. After careful consideration, we feel that it has merit but does not fully meet PLOS Global Public Health’s publication criteria as it currently stands. Therefore, we invite you to submit a revised version of the manuscript that addresses the points raised during the review process.

We look forward to receiving your revised manuscript.

Kind regards,

Kathleen Bachynski, PhD, MPH

Academic Editor

Journal Requirements:

Additional Editor Comments (if provided):

Thank you very much for submitting your manuscript to PLOS Global Health. The manuscript addresses an important area of research in global health. Both reviewers recommend revisions to further strengthen the manuscript, such as revising the presentation of the results for greater clarity. Reviewer #1 also recommended a number of additional points of clarification in the introduction, methods section (such as further details on the questionnaire used), and rewriting the study limitations section. Overall, revisions throughout the manuscript are needed to ensure that the study aims, discussion and conclusion are all thoroughly aligned. Therefore, I invite you to respond to the reviewers’ comments and revise your manuscript.

Reviewers' comments:

Reviewer's Responses to Questions

**Comments to the Author**

1. Does this manuscript meet PLOS Global Public Health’s publication criteria? Is the manuscript technically sound, and do the data support the conclusions? The manuscript must describe methodologically and ethically rigorous research with conclusions that are appropriately drawn based on the data presented.

Reviewer #1: Partly

Reviewer #2: Yes

2. Has the statistical analysis been performed appropriately and rigorously?

Reviewer #1: No

Reviewer #2: Yes

3. Have the authors made all data underlying the findings in their manuscript fully available (please refer to the Data Availability Statement at the start of the manuscript PDF file)?

Reviewer #1: No

Reviewer #2: Yes

4. Is the manuscript presented in an intelligible fashion and written in standard English?

Reviewer #1: Yes

Reviewer #2: Yes

5. Review Comments to the Author

Reviewer #1: This manuscript concerns welders and their working situation, and this is important. There are many welders in small informal companies, and the health and safety among these needs to be focused upon. The manuscript is interesting and large parts of it are important. However, some improvements are needed:

-The introduction is very nice written but does not tell us why particulate matter and noise are factors chosen for this study. These factors are interesting, but not the most important issues among welders. Welders are well known to have problems with running eyes due to lack of eye protection, and also to have burns/injuries. Focusing on noise and particles does not seem logical.

The aim of the study is unclear, as it is mentioned to see if the use of PPE improves health outcomes and reduces injury on job sites. There are no examinations of health and injuries in this manuscript. The aim should be revised.

Also, PPE can be many types, and the types needed for welders should have been described in the introduction.

-In methods, a questionnaire is mentioned briefly. It seems like the authors have not used any validated tool? The questionnaire needs to be described in detail if there is no reference the authors can give us that has these details described. Composite scores are fine, but we need to know what the scores actually includes. Also, it is not described how the questionnaire was used. Did the workers answer them on paper, or were they interviewed? And by whom and where did this happen? Method details are missing here.

A predefined checklist is also mentioned. It is such a good idea to use this method, but we need to know what the content of this was and how it was handled as a source for data. Types of PPE are listed, butt his is not sufficient. It is not always necessary with all types of PPE, this depends on the work done – therefore the authors need to describe how this observation was done. It is all very unclear.

Air and noise pollution is shortly mentioned – but this description is not detailed enough, the reader cannot grasp the validity of this part of the study. Maybe the best idea for this study is to delete everything concerning these factors and concentrate on the interviews and observational study?

As for statistics, it is difficult to evaluate if the methods chosen are the best, since the data are insufficiently described.

The results table 1 shows that jobs performed were several? This should be explained. The figures are impossible to understand, e.g. there are 219 participants. 73 work with welding? And 100 work with all activities? What are all activities? And is this their daily work? Or the work they did on the day of the interview? Lack of details on the wording if questions and why the questions were asked makes table 1 difficult to read.

At the end of 3.1, we can read that the participants had frequently reported illnesses? Where did this come from, which questions? And what is respiratory complications? This should be removed from the study, it is not information well obtained – at least it is not possible to grasp how it was asked about.

Statistical models are great, but a simple descriptions of the data are needed to understand the analyses. Which PPE did the welders use, and how many used what? To refer to scores is not sufficient for the reader to understand the situation on these workplaces. Simple descriptions should be done before the statistical regression analyses.

Table 2 and 3 and all the plots are not useful.

The result pages should be changed from text to tables. It is very difficult to see the results written like this in a text. This is especially so because we do not understand where the different variables came from, if it is from the questions or the observations. This needs to be totally rewritten and much simplified, and it is suggested to delete all about the noise and particles.

The discussion also needs to be revised if the previous text is changed.

The study limitations seem strange and must be wrong and belong to another study perhaps? The health issues have been unclear in the whole manuscript and is here very confusing. The limitations must include the weaknesses in the methods, it seems you used not validated questions for instance. Selection of worker must be discussed.

The conclusion must be aligned with the aims of the study; but these must be made clear at the start.

Hopefully the authors can improve this manuscript, as much of the information is of interest for the working life. But the revision is clearly needed – good quality of the manuscript is needed to give impact of the results.

Reviewer #2: Comment 1

On the title, I suggest you say ‘Occupational’ not occupation (Line 15).

Comment 2

On the Abstract, I suggest you clearly state your objectives for instance numbering them for easy flow (Line 19 to Line 22).

Comment 3

On the Introduction, I suggest you insert some sources of your assertion. You can make use of sites such as International Labour Organization etc (Line 51).

Comment 4

I suggest you cite sources, at least two to affirm your statement on Line 68 to Line 70. Cite sources from PLOS Global Health Journal or any other reputable institutions such as the World Health Organisation.

Comment 5

I suggest you revisit Line 180 to Line 190

Comment 2

I suggest you in your results, you include not just tables but bar graphs and pie charts

6. PLOS authors have the option to publish the peer review history of their article (what does this mean?). If published, this will include your full peer review and any attached files.

**Do you want your identity to be public for this peer review?** For information about this choice, including consent withdrawal, please see our Privacy Policy.

Reviewer #1: No

Reviewer #2: No

---

## [Decision Letter · Decision Letter 1]

29 Jan 2024

Informal welders’ occupational safety and environmental health risks in northwestern Tanzania

PGPH-D-23-01493R1

Dear Dr. Nyanza,

We are pleased to inform you that your manuscript 'Informal welders’ occupational safety and environmental health risks in northwestern Tanzania' has been provisionally accepted for publication in PLOS Global Public Health.

Best regards,

Kathleen Bachynski, PhD, MPH

Academic Editor

Reviewer Comments (if any, and for reference):

Reviewer's Responses to Questions

**Comments to the Author**

1. If the authors have adequately addressed your comments raised in a previous round of review and you feel that this manuscript is now acceptable for publication, you may indicate that here to bypass the “Comments to the Author” section, enter your conflict of interest statement in the “Confidential to Editor” section, and submit your "Accept" recommendation.

Reviewer #1: All comments have been addressed

Reviewer #2: All comments have been addressed

2. Does this manuscript meet PLOS Global Public Health’s publication criteria? Is the manuscript technically sound, and do the data support the conclusions? The manuscript must describe methodologically and ethically rigorous research with conclusions that are appropriately drawn based on the data presented.

Reviewer #1: Yes

Reviewer #2: Yes

3. Has the statistical analysis been performed appropriately and rigorously?

Reviewer #1: Yes

Reviewer #2: Yes

4. Have the authors made all data underlying the findings in their manuscript fully available (please refer to the Data Availability Statement at the start of the manuscript PDF file)?

Reviewer #1: Yes

Reviewer #2: Yes

5. Is the manuscript presented in an intelligible fashion and written in standard English?

Reviewer #1: Yes

Reviewer #2: Yes

6. Review Comments to the Author

Reviewer #1: The authors have responded well to the review comments. Two minor details:

1.The last part of the Discussion 378-381, I suggest to delete this text : «Due to COVID-19 restrictions and the need for adequate social distancing, we could not conduct a lung functioning test to understand effects of air pollution, nor did we assess noise induced hearing loss. Further..» These details are not interesting, now when you have removed noise and particles from the study. Start this paragraph with Health effects and ….++

2. Make sure you have a figure text for the figure you added, I could not find it.

Reviewer #2: I have thoroughly read the revised article and believe in the best of my knowledge that the authors have addressed all the comments

7. PLOS authors have the option to publish the peer review history of their article (what does this mean?). If published, this will include your full peer review and any attached files.

**Do you want your identity to be public for this peer review?** For information about this choice, including consent withdrawal, please see our Privacy Policy.

Reviewer #1: **Yes: **Bente E. Moen

Reviewer #2: **Yes: **Elvis Chiboyiwa
